# Building customizable auto-luminescent luciferase-based reporters in plants

Arjun Khakhar[1,2], Colby G Starker[1,2], James C Chamness[1,2], Nayoung Lee[3], Sydney Stokke[1,2], Cecily Wang[1,2], Ryan Swanson[1,2], Furva Rizvi[1,2], Takato Imaizumi[3], Daniel F Voytas[1,2]*

[1]Department Genetics, Cell Biology, & Development, University of Minnesota, Minneapolis, United States; [2]Center for Precision Plant Genomics, University of Minnesota, St. Paul, United States; [3]Department of Biology, University of Washington, Seattle, United States

**Abstract** Bioluminescence is a powerful biological signal that scientists have repurposed as a reporter for gene expression in plants and animals. However, there are downsides associated with the need to provide a substrate to these reporters, including its high cost and non-uniform tissue penetration. In this work we reconstitute a fungal bioluminescence pathway (FBP) *in planta* using a composable toolbox of parts. We demonstrate that the FBP can create luminescence across various tissues in a broad range of plants without external substrate addition. We also show how our toolbox can be used to deploy the FBP *in planta* to build auto-luminescent reporters for the study of gene-expression and hormone fluxes. A low-cost imaging platform for gene expression profiling is also described. These experiments lay the groundwork for future construction of programmable auto-luminescent plant traits, such as light driven plant-pollinator interactions or light emitting plant-based sensors.

## Introduction

Bioluminescence is used by a diverse set of organisms to achieve a broad range of goals, such as attracting mates, scaring off predators and recruiting other creatures to spread spores (*Shimomura, 2006*; *Wainwright and Longo, 2017*; *Verdes and Gruber, 2017*; *Labella et al., 2017*; *Oliveira et al., 2015*). The mechanism of light emission is broadly conserved: an enzymatic oxidation reaction by a luciferase enzyme turns a luciferin substrate into a high energy intermediate, which decays to produce light (*Shimomura, 2006*). Several different luciferin substrate-luciferase enzyme pairs have been described to date (*Oba et al., 2017*; *Schultz et al., 2018*). Researchers have leveraged some enzymes, such as the firefly and Renilla luciferases, to build reporters to study gene expression in plants and other eukaryotic systems (*Khakhar et al., 2018*; *Wend et al., 2013*). These reporters have a high signal to noise ratio because plants produce effectively no background bioluminescence signal.

Additionally, thanks to its water-soluble luciferin substrate, firefly luciferase can be used to visualize dynamic changes in gene expression in plants in a non-invasive manner, theoretically enabling whole-plant time lapse imaging (*Khakhar et al., 2018*). However, current bioluminescent reporters do present some significant challenges, which has limited their broad application for macro scale visualization of gene expression. One major challenge is uniform delivery and penetration of the luciferin substrate, especially in adult plant tissues. Some luciferins, such as coelenterazine – the substrate for Renilla luciferase – are non-water soluble and cannot be used for imaging without cell-lysis (*Wend et al., 2013*). The luciferin for firefly luciferase, D-luciferin, is water soluble and can be topically applied to whole plants or delivered through watering (*Rellán-Álvarez et al., 2015*). However, uniform substrate delivery is challenging to achieve, making it difficult to disambiguate whether an

*For correspondence:
voytas@umn.edu

Competing interests: The authors declare that no competing interests exist.

**eLife digest** Many animals have evolved the capacity to produce light from chemical reactions. For example, an enzyme known as luciferase in fireflies produces light by acting on a molecule called luciferin.

Scientists have identified the enzymes that drive several of these systems and used them to build reporters that can study the activity of genes in the tissues of plants and other lifeforms over space and time. However, these reporters often require chemicals to be added to the tissues to produce light. These chemicals tend to be expensive and may not penetrate evenly into the tissues of interest, limiting the potential applications of the reporters in research studies.

Recently, it has been discovered that fungi have a bioluminescence pathway that converts a molecule known as caffeic acid into luciferin. Caffeic acid is a common molecule in plants, therefore, it is possible the fungal bioluminescence pathway could be used to build reporters that produce light without needing the addition of chemicals.

Now, Khakhar et al. have inserted the genes that encode the enzymes of the fungal bioluminescence pathway into tobacco plants. The experiments found that this was sufficient to turn caffeic acid into molecules of luciferin which are able to produce light. Inserting the same genes into several other plant species, including tomatoes and dahlias, produced similar results. Further experiments showed that the fungal bioluminescence pathway can be used to build reporters that monitor the activity of plant genes throughout living tissues and over a period of several days as well as examine the response to plant hormones.

Alongside studying the activities of genes in plants, Khakhar et al. propose that the toolkit developed in this work could be used to generate plants with luminescence that can be switched on or off as desired. This could have many uses including helping plants attract insects to pollinate flowers and building plant biosensors that emit light in response to environmental signals.

absence of bioluminescent signal is due to low luciferase expression or poor substrate delivery. Additionally, the relatively high cost of D-luciferin (up to several hundred dollars per gram) makes continuous whole-plant imaging a prohibitively expensive practice.

To avoid the need to deliver the luciferin substrate, an optimal solution would be to engineer plants such that the substrate is produced in every cell. Krichevsky et al. demonstrated one approach to achieve this by incorporating the lux operon from the prokaryote *Photobacterium leiognathi* into the chloroplast genome in *Nicotiana tabacum* (**Krichevsky et al., 2010**). While auto-luminescence was observed, this approach is extremely challenging to extend to other plants due to the technical challenges of plastid transformation. Plastid localization is necessary, as the luciferin substrate produced by this pathway is toxic in some eukaryotic systems when nuclearly encoded (**Hollis et al., 2001**). In their 2018 paper, Kotlobay et al. identified a fungal bioluminescence pathway (FBP) consisting of a set of three genes sufficient to convert caffeic acid, a common plant metabolite (**Kotlobay et al., 2018**), into a luciferin molecule. The paired luciferase, *Luz*, can oxidize the luciferin and generate a bioluminescent signal (**Kotlobay et al., 2018**). This pathway can be nuclearly encoded, overcoming the challenges associated with plastid transformation. Additionally, the luciferase produces light in the green spectrum, which minimizes absorption by the predominant plant pigment, chlorophyll (**Kaskova et al., 2017**). This also makes it spectrally separable from the other major luciferase-based reporters, such as firefly luciferase, enabling the parallel deployment of both these reporters (**Kaskova et al., 2017**).

In this report we sought to reconstitute the FBP in planta and to test if natively produced caffeic acid could be channeled to generate bioluminescence. We also tested whether the FBP functioned across plant species, and if it could be used to study spatiotemporal patterns of gene expression. Further, we describe a toolkit of reagents to easily generate FBP-based reporters. We demonstrate how this toolkit can be deployed to generate programmable auto-luminescence patterns in planta and to build plant-based biosensors with luminescent outputs that do not require external substrate addition.

## Results

### Reconstitution of the fungal bioluminescence pathway in planta

To test if the fungal bioluminescence pathway described by *Kotlobay et al. (2018)* would function in planta, we attempted to transiently reconstitute this pathway in the leaves of *Nicotiana benthamiana*. We chose *N. benthamiana* because *Agrobacterium* infiltration provides an easy way to transiently express genes. We synthesized codon-optimized versions of the three genes necessary to convert caffeic acid into 3-hydroxyhispidin: 4'-phosphopantetheinyl transferase (*NPGA*) from *Aspergillus nidulans*, hispidin-3-hydroxylase (*H3H*) from *Neonothopanus nambi*, and a polyketide synthase (*Hisps*) also from *N. nambi* (*Kotlobay et al., 2018*; *Figure 1A*). We also generated a codon optimized version of the fungal luciferase, *Luz*. These DNA sequences were domesticated to remove commonly used type II-S restriction enzyme recognition sites, and made compatible with the MoClo plasmid assembly kit (*Engler et al., 2014*). This kit enables rapid swapping of promoters and terminators to create custom expression cassettes, which can be easily assembled into a single T-DNA with a one-step GoldenGate reaction. Together, our new components comprise a toolbox of vectors to construct FBPs for constitutive or spatio-temporally regulated auto-luminescence (*Table 1*).

When *N. benthamiana* leaves were imaged three days after delivery of the FBP via Agrobacterium infiltration, significant bioluminescence was observed over background at the site of infiltration (*Figure 1B,C,D*). This demonstrates the pathway can produce bioluminescence using natively synthesized caffeic acid. Comparing luminescence from leaves with the complete pathway versus controls with missing pathway enzymes, a bioluminescence signal was only observed with the complete pathway (*Figure 1—figure supplement 1D*). This indicates no native enzymes are expressed at functional levels in *N. benthamiana* leaves that could complement the functions of the FBP enzymes absent in these incomplete pathways. We also observed no bioluminescence from confluent cultures of Agrobacterium carrying the pathway (*Figure 1—figure supplement 1*), implying the observed bioluminescent signal was not from expression of this pathway in Agrobacterium.

### Incorporation of spent luciferin recycling pathway prolongs transient auto-luminescence

In their paper, (*Kotlobay et al., 2018*). also describe an additional enzyme, caffeylpyruvate hydrolase (*CPH*), which recycles the spent luciferin, caffeylpyruvic acid, back to the substrate for the pathway, caffeic acid (*Figure 1A*). We reasoned that incorporating this enzyme into our reconstituted pathway would create a stronger bioluminescence signal, due to a higher substrate concentration, and potentially prolong the presence of transient bioluminescence from Agrobacterium infiltration. To test this hypothesis, a new pathway was built with all the enzymes described above and an additional expression cassette for *CPH*. When we compared luminescence from Agrobacterium infiltrated *N. benthamiana* leaves, we observed approximately equal signals at four days, consistent with initial saturating substrate conditions (*Figure 1—figure supplement 2*). At later time points we saw higher mean luminescence signal from leaves infiltrated with the *CPH* containing pathway, consistent with the hypothesis that when the substrate is limited, the presence of a recycling pathway improves the bioluminescence signal (*Figure 1E*). Based on these results, all further experiments used FBPs including the recycling enzyme, *CPH*.

### Stable integration of the fungal bioluminescence pathway creates auto-luminescent plants

Our experiments thus far implied that auto-luminescent plants could be created by stably integrating expression cassettes for the three enzymes in the biosynthesis pathway, *NPGA*, *H3H*, and *Hisps*, the recycling pathway, *CPH*, and the fungal luciferase, *Luz*. We used the FBP characterized in *Figure 1* (FBP-6) to generate a stable transgenic line of *N. benthamiana*. When luminescence was characterized in plants on rooting medium (*Figure 1—figure supplement 4A*) or soil (*Figure 1F,G*, *Figure 1—figure supplement 4B*), we observed auto-luminescence throughout the plant. Stronger signals were observed in the root tips and shoot apical meristem, consistent with a higher density of cells in these tissues. Younger leaves also seemed to have lower luminescence than older leaves, indicating there is some developmentally consistent heterogeneity in luciferin substrate biosynthesis,

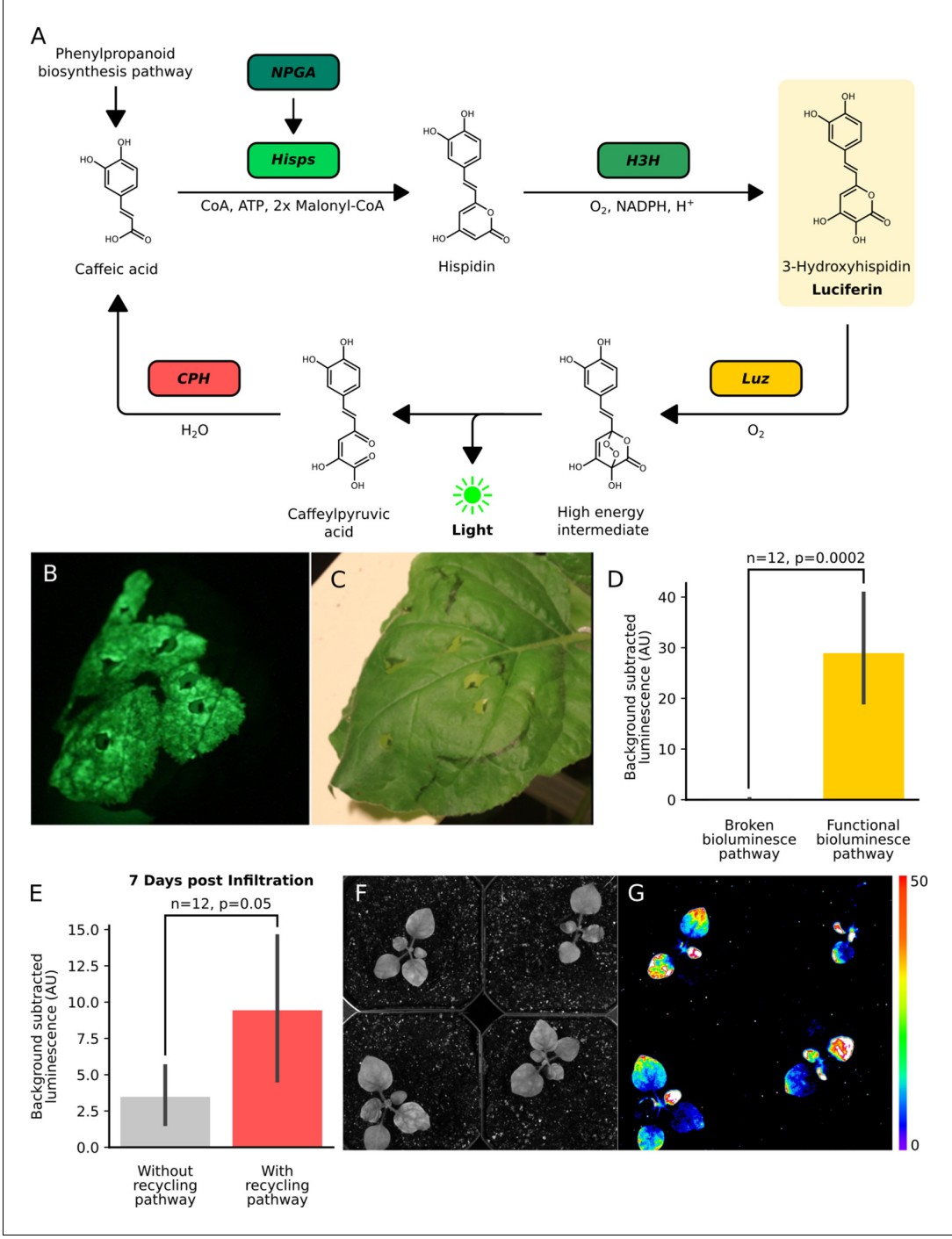

**Figure 1.** The fungal bioluminescence pathway creates auto-luminescence when transiently or stably expressed. (A) Schematic of the chemical reactions driving the generation of auto-luminescence in planta. Caffeic acid produced by the phenylpropanoid biosynthesis pathway is converted to Hispidin by *Hisps* once it is post-translationally activated by *NPGA*. Hispidin is then converted to 3-Hydroxyhispidin, the luciferin molecule, by *H3H*. Finally, the luciferase *Luz* oxidizes 3-Hydroxyhispidin to a high energy intermediate which degrades into Caffeylpyruvic acid, producing light. *CPH* can turn Caffeylpyruvic acid back into Caffeic acid, closing the cycle. (B) An image in the dark with an eight minute exposure of a *N. benthamiana* leaf infiltrated with the FBP demonstrating auto-luminescence in the infiltrated zone. (C) An image in the light of the same leaf with the infiltrated zone marked with a black outline. (D) Bar plots representing background subtracted luminescence from *N. benthamiana* leaves infiltrated with either a functional FBP (yellow) or a broken control (gray) missing the luciferase, *Luz*, three days after infiltration (n = 12, p=0.0002 based on a T-test). Black bars represent standard

*Figure 1 continued on next page*

*Figure 1 continued*

deviation. (E) Bar plots representing background subtracted luminescence seven days after infiltration from *N. benthamiana* leaves infiltrated with FBPs that either have (pink) or do not have (gray) the *CPH*-based recycling pathway (n = 12, p=0.05 based on a T-test). Black bars represent standard deviation. (F, G) Bright field image (F) and luminescence signal (G) captured with a CCD camera of transgenic *N. benthamiana* plants with a stable integration of the FBP (FBP-6) into the genome. Warmer colors correspond to higher luminescence in accordance with the lookup table.

The online version of this article includes the following source data and figure supplement(s) for figure 1:

**Source data 1.** Raw data for *Figure 1—figure supplements 1* and *2*.
**Source data 2.** Raw data for *Figure 1—figure supplements 1* and *2*.
**Source data 3.** Raw data for *Figure 1—figure supplement 3*.
**Figure supplement 1.** Characterization of luminescence from transient expression of different FBP pathway variants.
**Figure supplement 2.** Inclusion of a caffeic acid recycling pathway in the FBP prolongs luminescence signal.
**Figure supplement 3.** Co-expression of the Caffeic acid biosynthesis pathway with the FBP increases luminescence signal.
**Figure supplement 4.** Stable integration of the FBP into the genome results in the creation of auto-luminescent plants.

potentially due to variation in caffeic acid availability (*Figure 1F,G*). These results demonstrate it is possible to create sustained auto-luminescence by generating plants with stably integrated FPBs.

## Expression of a caffeic acid biosynthesis pathway increases luminescence

Caffeic acid is a requisite intermediate of lignin biosynthesis and is thus ubiquitous in higher plants (*Davin and Lewis, 1992*). However, caffeic acid levels may vary between tissues and under different environmental conditions (*Mitiouchkina et al., 2019*) This variation could complicate deployment of FBP reporters for gene expression. One solution might be to constitutively express enzymes that synthesize caffeic acid from a widely available plant metabolite. To test this, we built a T-DNA to constitutively express the *Rhodobacter capsulatus* tyrosine ammonia lyase and two *Escherichia coli* 4-hydroxyphenylacetate 3-monooxygenase components (HpaB, HpaC). Together these enzymes are known to catalyze caffeic acid synthesis from tyrosine (*Kotlobay et al., 2018*). Upon infiltration of this construct into leaves of a *N. benthamiana* line stably expressing the FBP, a significant increase in bioluminescence signal was observed compared to empty vector controls infiltrated in parallel on different sections of the same leaf (*Figure 1—figure supplement 3*).

## Auto-luminescence in different plant species

While *N. benthamiana* is an excellent model for prototyping, FBP should work in any plant species as long as appropriate promoters and terminators are used to express the pathway enzymes and as caffeic acid is ubiquitous in higher plants (*Davin and Lewis, 1992*). To test this hypothesis, we tested the FBP in multiple plant species that were amenable to Agrobacterium infiltration. We performed transient Agrobacterium-based delivery of the FBP into the model plant *Arabidopsis thaliana* and the crop plant *Solanum lycopersicum* (tomato) (*Wu et al., 2014*). We observed robust auto-luminescence signal over background in the cotyledons of treated seedlings (*Figure 2A,B*). We also performed Agrobacterium infiltrations of the leaves of the ornamental plant, Dahlia, and again observed auto-luminescence (*Figure 2C*). Finally, we explored if the FBP was functional in petals, as luminescence in this tissue has future applications in floriculture as well as for engineering plant-pollinator interactions. Agrobacterium infiltrations with the FBP of petals from *Catharanthus roseus* (periwinkle), *Petunia hybrida*, and three different varieties of *Rosa rubiginosa* (Rose) resulted in auto-luminescence in all flowers with a range of intensities (*Figure 2D,E*, *Figure 3A*). We also observed that the signal dropped off within a day of the flowers being detached from the plant body, whereas it persisted when the flowers remained connected to the plant body. This might mean some of the pathway substrates or co-factors are trafficked from source tissues; however, more tests are required to confirm this hypothesis. These results demonstrate that the FBP can function across a wide range of plants.

**Table 1.** Vectors containing FBP components.

| Plasmid Name | Fungal Bioluminescent Pathway Components | Plasmid Type |
|---|---|---|
| P303-FBP_2 | p35S_Short_TMV 5':AsNpga:tAtug7, pCmYLCV:NnH3H:AtuOCS, pCmYLCV:NnHisps:tHsp, p35S:NnLuz:tAtuOcs | T-DNA |
| P307-FBP_6 | pLHB1B1:NnCPH:tSlH4, p35S_Short_TMV 5':AsNpga:tAtug7, pCmYLCV:NnH3H:AtuOcs, pCmYLCV:NnHisps:tHsp, p35S:NnLuz:tAtuOcs | T-DNA |
| P313-FBP_8 | pLHB1B1:NnCPH:tSlH4, p35S_Short_TMV 5':AsNpga:tAtug7, pCmYLCV:NnHisps:tHsp, pAtuMas:Nn Luz:trbcSE9 | T-DNA |
| P315-FBP_10 | pUBQ1:NhH3H:tNos, pCmYLCV:NnHisps:tHsp, pAtuMas:NnLuz:trbcSE9 | T-DNA |
| P336-FBP_11 | pLHB1B1:NnCPH:tSlH4, p35S_Short_TMV 5':AsNPGA:tAtug7, pCmYLCV:NnH3H:AtuOcs, pCmYLCV:NnHisps:tHsp | T-DNA |
| P362-FBP_12 | pSlH4:NnCPH:tSlH4, p35S_Short_TMV 5':AsNpga:tAtug7, pAtuMas:NhH3H:tAtuMas, p35S:NnHisps:t35S, p35S:NnLuz:AtuOcs | T-DNA |
| P427-FBP_13 | pSlH4:NnCPH:tSlH4, pSlRBCS2:AsNpga:tAtug7, pAtuMas:NhH3H:tAtuMas, p35S:NnHisps:t35S, pAtELF4:NnLuz:trbcse9 | T-DNA |
| P445-FBP_17 | pSlH4:NnCPH:tSlH4, pSlRBCS2:AsNpga:tAtug7, pAtuMas:NnH3H:tAtuMas, p35S:NnHisps:t35S, pPhODO1:NnLuz:tAtuOcs | T-DNA |
| P447-FBP_19 | pSlH4:NnCPH:tSlH4, pSlRBCS2:AsNpga:tAtug7, pAtuMas:NnH3H:tAtuMas, p35S:NnHisps:t35S, pAtRAB18:NnLuz:tAtuOcs | T-DNA |
| P532 | 35S Short_TMV 5':HpaB:tAtug7, 35S Short_TMV 5':HpaC:tAtug7, 35S Short_TMV 5':RcTAL:tAtug7 | T-DNA |
| P195 | pMOD_A''_p35S_Short_TMV 5':AsNpga:tAtug7 | Module *Čermák et al., 2017* |
| P212 | pMOD_D_p35S:NnLuz:tAtuOcs | Module *Čermák et al., 2017* |
| P223 | pMOD_A'_pLHB1B1:NnCPH:tSlH4 | Module *Čermák et al., 2017* |
| P364 | pMOD_A'_pSlH4:NnCPH:tSlH4 | Module *Čermák et al., 2017* |
| P366 | pMOD_A''_pSlRBCS2:AsNpga:tAtug7 | Module *Čermák et al., 2017* |
| P368 | pMOD_B_pAtuMas:NnH3H:tAtuMas | Module *Čermák et al., 2017* |
| P440 | pMOD_D_pPhODO1:NnLuz:tAtuOcs | Module *Čermák et al., 2017* |
| P442 | pMOD_D_pAtRAB18:NnLuz:tAtuOcs | Module *Čermák et al., 2017* |
| pCO486 | pMOD_A_35S Short_TMV 5':AsNpga:tAtug7 | Module *Čermák et al., 2017* |
| pCO487 | pMOD_B_CmYLCV:NnH3H:tAtuOcs | Module *Čermák et al., 2017* |
| pCO488 | pMOD_C'_CmYLCV:NnHisps:tHsp | Module *Čermák et al., 2017* |
| pCO489 | pMOD_D_AtuMas:NnLuz:trbcSE9 | Module *Čermák et al., 2017* |
| pCO601 | pMOD_C'_p35S:NnHisps:t35S | Module *Čermák et al., 2017* |
| P525 | pMOD_A_35S Short_TMV 5':HpaB:tAtug7 | Module *Čermák et al., 2017* |
| P526 | pMOD_B_35S Short_TMV 5':HpaC:tAtug7 | Module *Čermák et al., 2017* |
| P527 | pMOD_C_35S Short_TMV 5':RcTAL:tAtug7 | Module *Čermák et al., 2017* |
| pCO600 | NnHisps | MoClo Part *Engler et al., 2014* |

*Table 1 continued on next page*

*Table 1 continued*

| Plasmid Name | Fungal Bioluminescent Pathway Components | Plasmid Type |
| --- | --- | --- |
| pCO602 | NnCPH | MoClo Part *Engler et al., 2014* |
| pCO603 | NnLuz | MoClo Part *Engler et al., 2014* |
| pCO604 | NnH3H | MoClo Part *Engler et al., 2014* |
| pCO605 | AsNPGA | MoClo Part *Engler et al., 2014* |
| pCO706 | hpaB | MoClo Part *Engler et al., 2014* |
| pCO707 | hpaC | MoClo Part *Engler et al., 2014* |
| pCO708 | RcTAL | MoClo Part *Engler et al., 2014* |

## Visualizing spatio-temporal patterns of gene expression

In planta substrate-independent bioluminescence promises to be a powerful reporter technology, as it enables long-term characterization of gene expression on a macro scale and in a cost-effective manner. The approach also does not suffer from the issues associated with current bioluminescent reporters, including substrate application and non-uniform substrate penetration. To validate that the FBP can be used to study spatio-temporal patterns of gene expression, we built a pathway variant in which *Luz* expression was driven by the promoter of the *ODORANT1* (*ODO1*) gene from petunia (*Fenske et al., 2015*; *Van Moerkercke et al., 2011*; *Verdonk et al., 2005*; *Figure 3B*). This gene has been previously characterized in petunia with a firefly luciferase reporter, where its expression was observed only in the flowers. *ODO1* expression was also shown to be diurnal, peaking in the evening at the transition of day to night (*Fenske et al., 2015*). We used Agrobacterium infiltration to deliver two versions of FBP to petunia flower petals: one with *Luz* expressed from the *ODO1* promoter (pODO1), and the other with *Luz* expressed from the constitutive 35S promoter. We then performed time course imaging of detached flowers. In the flowers treated with the pODO1:*Luz* we observed an increase in luminescence in the evening at the transition between day and night, a trend not seen in the 35S:*Luz* control (*Figure 3A,C*).

Visualizing multiple diurnal oscillations in flowers is technically challenging, as the auto-luminescence signal only lasts approximately one day after the flower is separated from the plant body, a requirement for mounting in the imaging platform. To overcome this limitation, we infiltrated two sets of flowers 24 hr apart and then harvested tissue and began time course imaging on the same day. In this way we were able to quantify luminescence signal from the pODO1:*Luz* and p35:*Luz* constructs over a two-day period post-infiltration. We observed the expected luminescence signal peak at the transition between day and night both one and two days after infiltration (*Figure 3C*), with a diminishing signal over time, which can be explained by the gradual loss of T-DNA expression over time (*Norkunas et al., 2018*) and gradual senescence of the detached flower. In contrast, the p35S:*Luz* controls showed an initial increase over time until maximal T-DNA expression was reached – crucially, not at the day-night transition – followed by a steady decrease, again attributed to T-DNA loss and senescence (*Figure 3C*). When we plot the time at which we observe peak luminescence signal in these time course experiments, we observe that the signal from the pODO1:*Luz* constructs peak around the expected time, at the transition of day to night, in contrast to the p35S controls, which do not (*Figure 3D*). At 1 day post infiltration this trend is visible but not significant, potentially due to the initial variability in Agrobacterium delivery and associated expression of the delivered genes, but by two days post infection we see a significant difference between the time of peak signal of pODO1:*Luz* and the control (*Figure 3D*). This difference in the temporal pattern of luminescence between pODO1:*Luz* and p35S:*Luz* indicates that the observed diurnal oscillations are not due to natural variation in the availability of caffeic acid, and highlights the importance of running

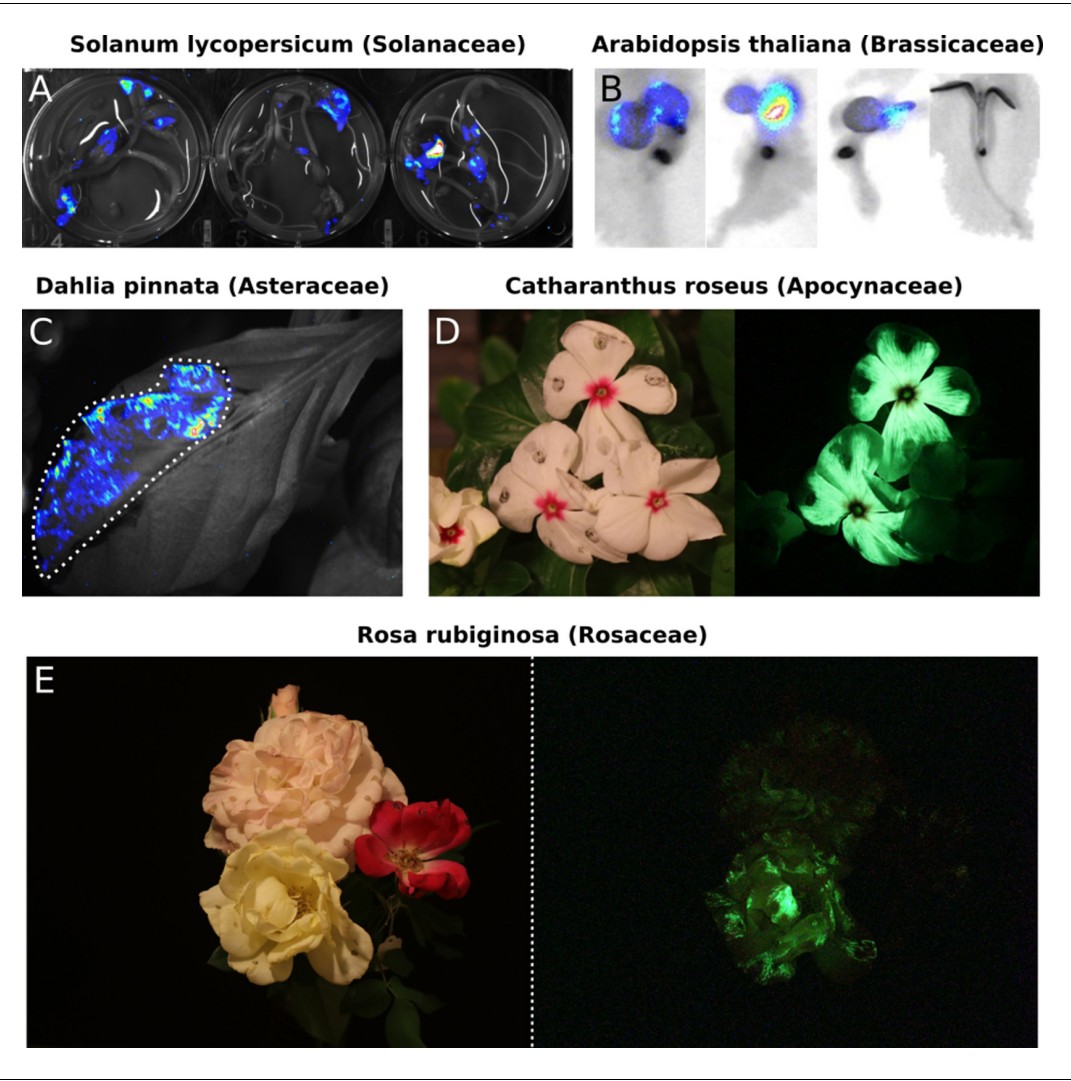

**Figure 2.** Transient expression of the FBP can generate auto-luminescence in a range of plant species.
(**A**) Luminescence signal superimposed on a bright field image of *Solanum lycopersium* (tomato) seedlings with an FBP delivered via the AgroBEST protocol. Warmer colors correspond to higher luminescence signal. (**B**) Luminescence signal superimposed on bright field images of *Arabidopsis thaliana* seedlings with a functional FBP (left three seedlings) or a broken FBP (right most seedling) delivered via the AgroBEST protocol. Warmer colors correspond to higher luminescence signal. (**C**) Luminescence signal superimposed on a bright field image of a *Dahlia pinnata* leaf infiltrated with an FBP. (**D**) Lit (left) and unlit, long exposure (right) true color images of *Catharanthus roseus* flowers infiltrated with an FBP. (**E**) Lit (left) and unlit, long exposure (right) true color images of *Rosa rubiginosa* flowers infiltrated with an FBP.
The online version of this article includes the following figure supplement(s) for figure 2:

**Figure supplement 1.** DSLR-based imaging produces comparable images of FBP luminescence to CCD-based imaging systems.

parallel constitutive promoter controls for proper interpretation of FBP reporters. These results demonstrate that the FBP can be used in a similar fashion to firefly luciferase to study spatio-temporal patterns of gene expression, but without requiring substrate addition. The experiments also serve as proof-of-concept that, by using an appropriate set of promoters to drive the pathway genes, an FBP can be programmed to generate specific spatio-temporal patterns of auto-luminescence in stable transgenic lines for synthetic biology applications.

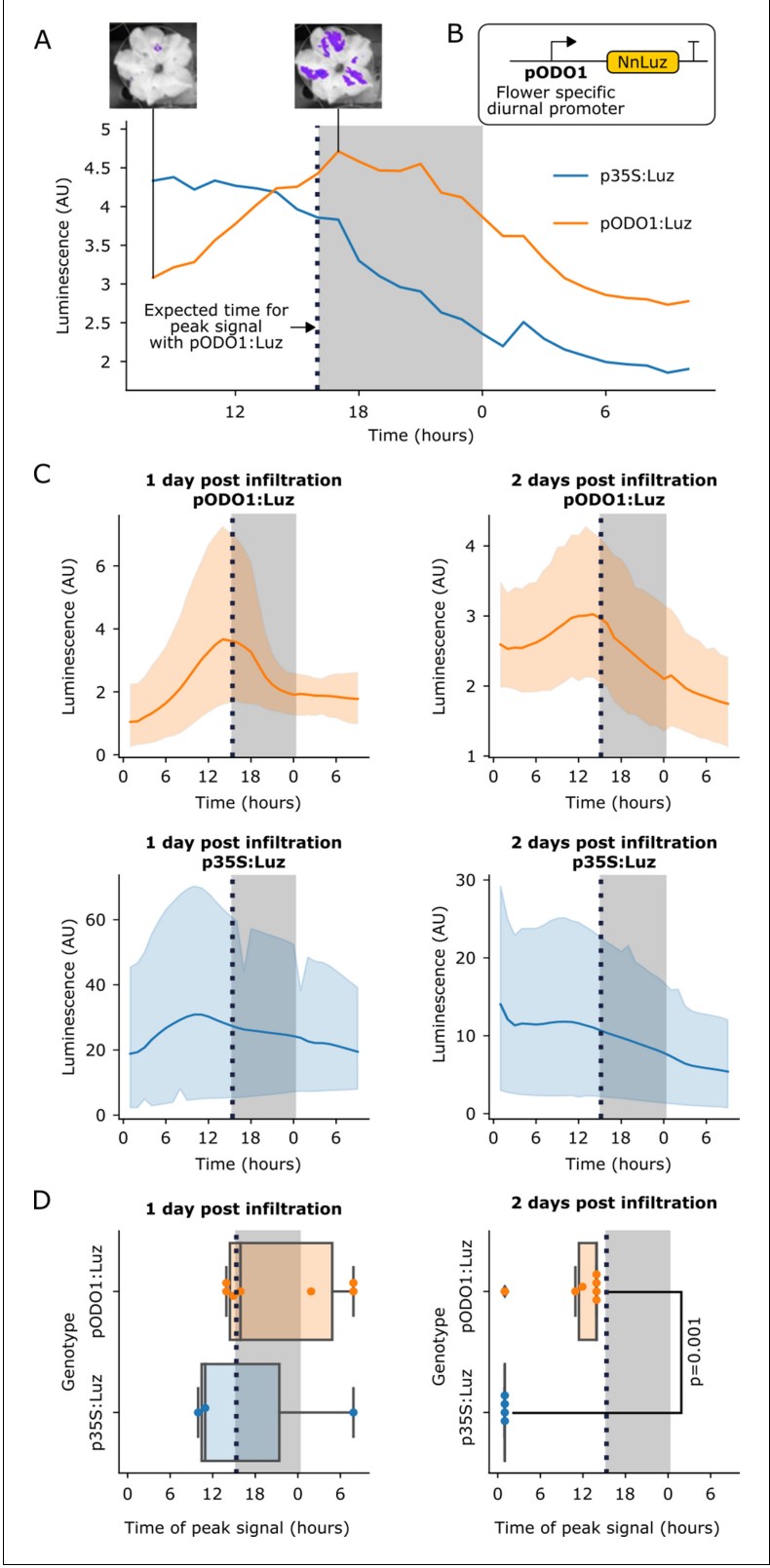

**Figure 3.** The FBP can be used to build reporters of the spatio-temporal patterns of gene expression. (**A**) A representative example of time course luminescence data from long day entrained *P. hybrida* flowers infiltrated with FBPs. The orange line represents FBP with *Luz* driven by the *ODO1* promoter from Petunia and the blue line represents the mean luminescence signal of an FBP with *Luz* driven by the 35S promoter. The gray background
*Figure 3 continued on next page*

*Figure 3 continued*

represents the night period. The dashed black line highlights where we expect to see the peak expression from the ODO1 promoter. Representative images of petunia flowers infiltrated with the pODO1:*Luz* at minimum and maximum bioluminescence are also shown, where luminescence signal is overlaid on the bright field image. (B) A schematic of the *Luz* expression cassette being driven by flower specific diurnal promoter of the petunia gene, *ODO1*, which is expected to have peak expression at the day to night transition. (C) Time course luminescence data from long day entrained *P. hybrida* flowers that were infiltrated with an FBP with *Luz* driven by the *ODO1* promoter, 1 and 2 days post infiltration (n = 7 for both) is shown in orange. The time course luminescence data from controls imaged in parallel that were infiltrated with an FBP with *Luz* driven by the *35S* promoter, 1 (n = 4) and 2 (n = 3) days post infiltration is also shown, in blue. The solid lines are mean luminescence and light outlines represent standard deviation. Time (hours) shows Zeitgeber time (time after the light onset at 9AM). The gray background represents the night period. The dashed black line highlights where we expect to see the peak expression from the ODO1 promoter. (D) Box plots summarizing when we observe peak luminescence signal in the time course luminescence data shown in C. Each dot represents an independent biological replicate of a flower infiltrated with a pathway that contains either pODO1:Luz (orange) or p35S:Luz (blue). The gray background represents the night period. The dashed black line highlights where we expect to see the peak expression from the ODO1 promoter. The reported p value was calculated based on a t-test.

The online version of this article includes the following source data for figure 3:

**Source data 1.** Raw data plotted in *Figure 3*.
**Source data 2.** Raw data plotted in *Figure 3*.
**Source data 3.** Raw data plotted in *Figure 3*.

## Visualizing hormone signaling dynamics in planta

Hormones coordinate diverse aspects of plant metabolism and development by carrying information from cell to cell across tissues and organ systems in the plant. Firefly luciferase reporters have been an invaluable tool to study how hormone fluxes can trigger developmental changes in plants (*Khakhar et al., 2018*). However, the challenges of substrate cost, application and penetration make using them across the entire plant body or over long periods problematic. The substrate independence of the FBP makes it an attractive alternative to overcome these challenges.

To demonstrate how to create a hormone reporter from FBP, we used the *AtRAB18* promoter (pRAB18) to drive expression of *Luz* (*Figure 4A*). This promoter has previously been characterized as driving a strong and selective expression increase in response to the plant hormone abscisic acid (ABA) (*Kim et al., 2011*). We first validated whether FBP could report ABA levels in a dose-dependent manner in planta. To do this, we assayed the bioluminescent signal produced three days after infiltration into *N. benthamiana* leaves that were treated with increasing concentrations of ABA. As expected, we observed increasing bioluminescence signal with increasing doses of ABA (*Figure 4B*). The sensitivity of the pRAB18 driven FBP reporter was consistent with the previously characterized pRAB18 driven GFP reporter, as both were responsive to 10 µM ABA (*Kim et al., 2011*).

We next tested whether this system could respond to endogenous ABA signals triggered by desiccation. One set of *N. benthamiana* plants was not watered for several days whereas another set was kept well-watered. The pRAB18 driven FBP was delivered to leaves of these plants via Agrobacterium infiltration. Leaves were then detached and imaged over time; however, leaves from the unwatered plants were allowed to desiccate further, while the other set was kept moist. We expected that as the leaves dried, they would begin to produce ABA (*Ramachandra Reddy et al., 2004*), leading to an increase in luminescence. In fact, we observed a significant increase in luminescence in desiccating leaves, up to levels of the unwatered p35S:*Luz* positive control (*Figure 4C*). We did not see a similar increase in signal from pRAB18 driven FBP in leaves that were kept moist (*Figure 4—figure supplement 1*).

We repeated the experiment without detaching the infiltrated leaves from the plant body and again observed an increase in luminescence in the leaf infiltrated with pRAB18 driven FBP on the unwatered plant over time, as compared to the watered plant control, which showed no increase in luminescence (*Figure 4D*, *Figure 4—video 1*). Luminescence from the p35S:*Luz* control remained similar for the watered and un-watered conditions (*Figure 4D*, *Figure 4—video 1*), demonstrating that the dynamics visualized by the pRAB18:*Luz* reporter are not due to variation in caffeic acid levels or differential expression patterns of other promoters in the pathway. These results demonstrate

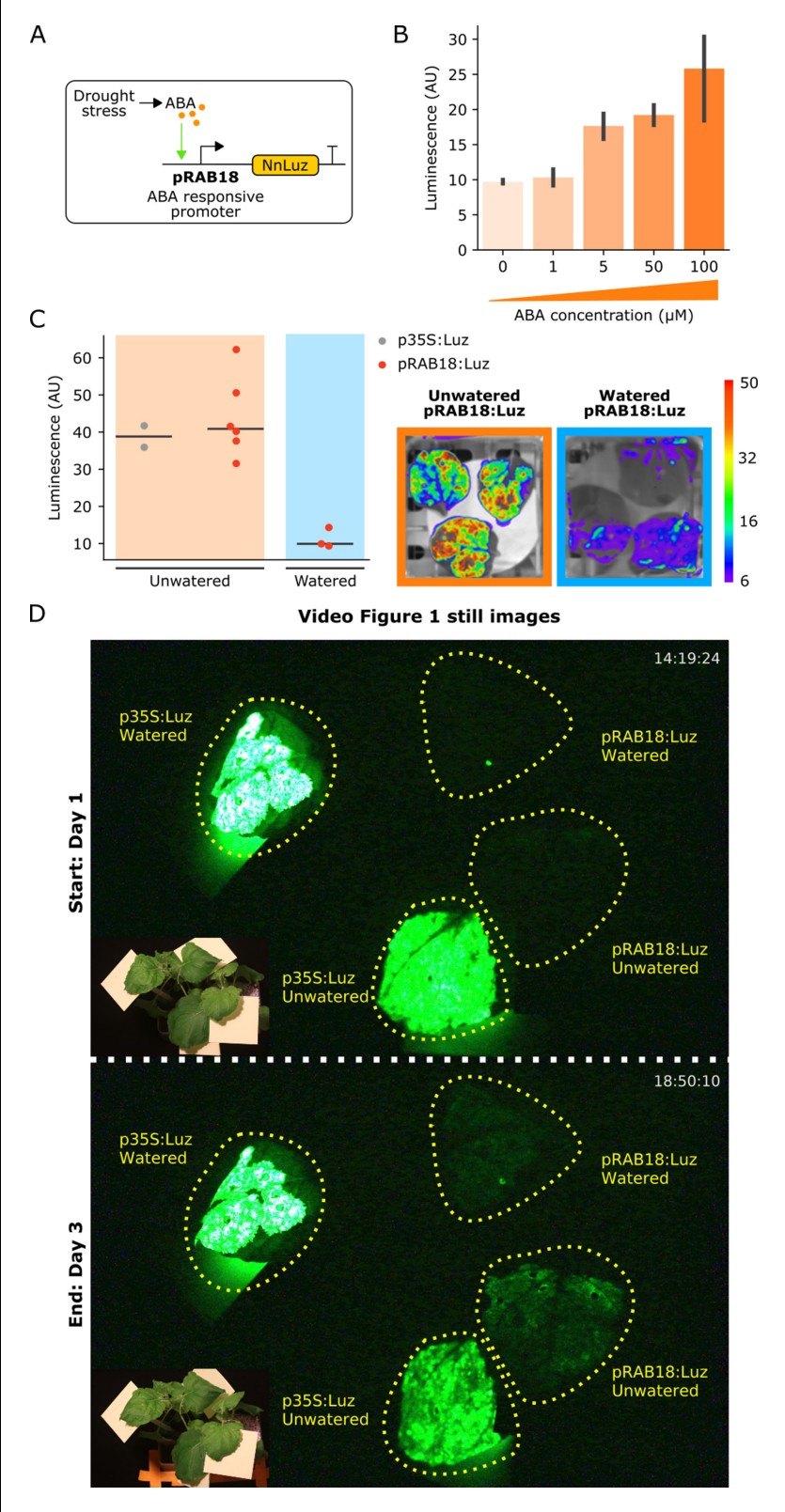

**Figure 4.** The FBP can be used to build reporters of hormone signaling dynamics *in planta.* (**A**) Schematic of the *Luz* expression cassette driven by the ABA-responsive *AtRAB18* promoter. (**B**) Bar plots summarizing mean luminescence signal observed from *N. benthamaina* leaves co-infiltrated with the pRAB18:*Luz* FBP variant along with increasing concentrations of the hormone ABA and then imaged after three days. Black bars represent

*Figure 4 continued on next page*

*Figure 4 continued*

standard deviation (n = 3). (**C**) Luminescence signal observed from *N. benthamaina* leaves infiltrated with an FBP and then either kept moist (blue background) or allowed to desiccate to trigger an ABA signal (orange background). Gray and red dots represent data from independent leaves infiltrated with FBP with a 35S or pRAB18 driven *Luz* respectively. Representative images of watered and un-watered leaves infiltrated with the pRAB18:*Luz* FBP with luminescence signal overlaid on a bright field picture. Warmer colors represent higher signals. (**D**) Still images from the start and end of Video *Figure 1*, where one leaf each on two *N. benthamiana* plants were agro-infiltrated with FBPs that had either a 35S or pRAB18 driven *Luz*. One plant was allowed to desiccate (bottom two leaves labeled unwatered) while the other was kept watered (top two leaves labeled watered). The infiltrated leaves are highlighted with a dashed yellow line. A paired bright field image is inset in corner of each image. The online version of this article includes the following video, source data, and figure supplement(s) for figure 4:

**Source data 1.** Raw data for *Figure 4*.
**Figure supplement 1.** An FBP with pRAB18 driving *Luz* expression shows increased luminescence in unwatered conditions.
**Figure 4—video 1.** Time lapse movie of *Nicotiana benthamiana* transiently expressing an FBP with pRAB18 driving *Luz* expression in watered and unwatered conditions with parallel positive controls.
https://elifesciences.org/articles/52786#fig4video1

the utility of the FBP as a tool to enable substrate independent visualization of hormone signaling in planta.

## Developing a low-cost, long-term bioluminescence imaging system

The power of bioluminescence as a tool to study dynamic signals in biology, such as ABA accumulation during drought, is challenging to access in lower-resource settings due to the high costs of substrate and instrumentation. The substrate independence of the FBP makes it more economical than other bioluminescent systems. However, the high cost and small chamber size of commercially available luminescence imaging setups make running multiple experiments in parallel cost prohibitive: such commercial systems can exceed $100,000 USD. To address this challenge, we leveraged a wealth of open-source software and off-the-shelf hardware components to design and build a low cost, modular platform for bioluminescence imaging. Our platform uses a Raspberry Pi single board computer as a controller for a DSLR camera, outfitted with a macro lens and mounted in either a dark room or pop-up plant growth tent. The code used for controlling customized still and time-course imaging with this platform, as well as the image processing pipeline, is available on Github

**Table 2.** Components for DSLR-Based Imaging Platform.

| Component | Source | Cost |
|---|---|---|
| Hardware | | |
| RaspberryPi 3 Complete Starter Kit | CanaKit | $75 |
| Canon EOS 80D DSLR | Canon | $1100 |
| Canon AC adapter and DC coupler | B and H Photo | $150 |
| EF 100 mm f/2.8 Macro Lens | Canon | $600 |
| Tripod | Amazon | $25 |
| Controllabe 4-Outlet Power Relay | Adafruit | $25 |
| Jumper cables | Amazon | $10 |
| Pop-up plant growing tents | Zazzy | $153 |
| Software | | |
| RPi.GPIO | https://pypi.org/project/RPi.GPIO/ | Free |
| gphoto2 | http://gphoto.org/ | Free |
| ImageJ | https://imagej.nih.gov/ij/ | Free |
| ffmpeg | https://ffmpeg.org/ | Free |
| Python wrappers | https://github.com/craftyKraken/P4 | |

(*Table 2*). The entire cost comes to less than $2,500, the bulk of which is for the camera and lens; substituting an entry-level DSLR and kit lens would bring the cost below $2000. Programmatic control of the setup can be conducted directly on the Pi, or from any networked computer.

With programmatically controlled long exposures and a tripod stabilizer, the DSLR camera provides high resolution and detail for macroscopic plant subjects under multiple light conditions, and sufficient sensitivity to capture the bioluminescence signal from stable or transient expression of the FBP. We do observe higher levels of thermodynamic noise than with cooled CCD camera setups, but this is easily filtered using open-source software such as ImageJ (*Schindelin et al., 2012*). For comparison, a side-by-side example is given in *Figure 2—figure supplement 1*). The same periwinkle petal infiltration was imaged under first the CCD camera (*Figure 2—figure supplement 1A*) and then the DSLR (*Figure 2—figure supplement 1B*). The DSLR captured high resolution, true color images under both the light and dark conditions, while showing similar light sensitivity to the reporter.

In contrast to all-in-one tabletop chamber systems, our setup with a dark room or pop-up tent (*Katagiri et al., 2015*) permits imaging of much larger plants and flexible positioning of the camera relative to the subject. For example, we took images of flower petal infiltrations on intact rose bushes a meter in height. These features make our platform a compelling alternative to commercially available systems at a fraction of the cost. We used this platform to capture high-resolution images of various plant subjects (*Figure 2B*, *Figure 2D,E*), and to perform time-course imaging for an FBP under an *AtRAB18* drought-inducible promoter (*Figure 4—video 1*). These results illustrate how our platform, in conjunction with the FBP, enable low cost bioluminescence reporting for dynamic biological phenomena.

## Discussion

In this report, we demonstrate that the FBP can convert a common plant metabolite, caffeic acid, into a luciferin and use it to produce robust luminescence in planta. We go on to show how the FBP is a useful addition to the bioluminescent reporters available for plants. Because it emits in the green spectrum (*Kaskova et al., 2017*), *Luz* should be spectrally separable from firefly luciferase, permitting parallel deployment. Our observation that incorporating the recycling pathway prolongs the luminescent signal suggests an avenue to enable long term imaging without substrate depletion in stable lines. This may also explain why we observed stronger luminescence in infiltrated tissue than in the stable transgenic line. However, this might also be due to overexpression of the enzymes from high-copy delivery of the T-DNAs in our transient leaf infiltration assay. Thus, further optimizing the expression of the enzymes in the FBP might be another avenue to increase the luminescence. Such re-engineering can easily be implemented with our FBP toolkit, thanks to modular design and assembly within the MoClo system (*Engler et al., 2014*).

Our observations from transient expression of a caffeic acid biosynthesis pathway in stable transgenic lines expressing the FBP highlight increasing the available caffeic acid in target tissues as another potential avenue for increasing the strength of auto-luminescence generated by the FBP. In the future this pathway might be used in conjunction with the FBP to create reporters that are more robust against natural variation in caffeic acid biosynthesis.

Transient expression assays demonstrate that the FBP functions in a broad range of plant species. With the recent development of rapid morphogen-mediated plant transgenesis protocols (*Lowe et al., 2018*), we believe FBP-based biosensors and reporters can be extended across a range of plants for long term visualization of gene expression. This extensibility to a broad range of plants makes this approach to auto-luminescence superior to the existing transplastomic approach (*Krichevsky et al., 2010*), as plastid transformation is technically challenging and currently limited to species such as tobacco and a few related *Solanaceae* (*Rigano et al., 2012*).

Our results with the pODO1 driven *Luz* in petunia flowers demonstrate how the FBP can be used to study the spatiotemporal patterns of gene expression, reporting the same patterns of gene expression as a firefly luciferase reporter (*Fenske et al., 2015*), but without the requirement for substrate addition. Compared to these previously published luciferase reporters, which are typically provided an excess of substrate during application, we observe a lower amplitude response with our FBP reporters, which rely on an endogenously synthesized luciferin substrate. This highlights the scope for additional reporter optimization by increasing the pool of available substrate, potentially

through manipulation of caffeic acid biosynthesis (*Figure 1—figure supplement 3*). Further characterization of the stability of the Luz protein will be needed to determine the temporal resolution of FBP-based reporters. However, the data presented here do show that our FBP reporters can be used to visualize changes in gene expression that occur on the timescale of hours. The development of stable FBP transgenic lines here and in parallel work by Mitiouchkina et al. demonstrate how our FBP tools could be applied in the future to visualize gene expression in various tissues over the lifetime of the plant (*Mitiouchkina et al., 2019*), and thus lower the cost of performing these types of experiments. Further, the data show how by swapping promoters in the pathway, spatio-temporal patterns of auto-luminescence can be programmed into plants for synthetic biology applications. In its natural context, the FBP is used by fungi to generate luminescence to lure insects that spread their spores (*Oliveira et al., 2015*). It may then be possible to engineer plants with sufficiently bright auto-luminescent flowers to attract nocturnal insects, and thereby drive novel plant-pollinator interactions. Bioluminescent flowers would also have relevance for designer floriculture, and might serve as a visually compelling demonstration of the power of synthetic biology to the public. We plan to generate stable FBP petunia lines and explore this avenue of research further in the future.

The data we present using the pRAB18-driven *Luz* to track drought stress via ABA signaling demonstrates how the FBP could be used to build biosensors for internal or external signals by driving genes in the pathway with synthetic promoters. In the future, stable lines generated with similar constructs could be deployed to monitor the environmental impact on plants at field scales. These biosensors would have an output more easily visualized than fluorescent reporters, have a high signal to noise ratio, and be cost effective due to their substrate independence. Thanks to the ease of promoter swapping in our FBP toolkit, this approach could be easily extended to build biosensors for a range of different phytohormones, environmental cues like light, or synthetic signaling systems for basic science and translational applications.

Besides the substrate costs, the high cost of commercially available bioluminescence imaging systems restrict access to luminescence-based reporters. We demonstrate how commercially available photography equipment and open-source software can be used to set up a relatively low-cost, modular and programmable luminescence imaging platform. Coupled with the substrate independence of the FBP, this will broaden access to these tools to lower resource settings, and enable scalable application of these reporters. We hope the work presented here will serve as a first step for the creation of novel bioluminescence-based tools in plants for basic science discovery and synthetic biology enabled applications.

## Materials and methods

### FBP plasmid construction

The various FBP-encoding T-DNA constructs characterized in this work were built using a two-step process. First, base plasmids containing an expression cassette were built for each of the five enzymes in the FBP. These base plasmids were designed with promoters for either constitutive or tissue/time-period specific expression of the FBP, sourced either from the MoClo toolkit (*Engler et al., 2014*) or amplified from genomic or plasmid DNA with primers adding the appropriate BsaI restriction enzyme recognition sites. The AtRAB18 and PhODO1 promoters were amplified from published plasmids obtained from either Addgene or via requests from the authors. Terminators were chosen from the MoClo kit to ensure high expression (*Engler et al., 2014*). DNA sequences encoding the enzymes were codon optimized for *N. benthamiana*. Additionally, all the common type-IIS restriction enzyme recognition sites were omitted from the synthesized molecules through synonymous base pair changes. These sequences were synthesized by Twist Biosciences. All these parts were assembled into the base vectors with a BsaI-based GoldenGate assembly reaction. The base vector backbones were designed to contain the appropriate AarI sites to be assembled together into either four or five expression cassettes containing T-DNAs through an AarI-based GoldenGate assembly reaction (*Čermák et al., 2017*). All vectors listed in *Table 1* are available via Addgene or upon request.

## Agrobacterium infiltration for transient expression of FBP

For all the transient expression assays of an FBP besides those shown in *Figure 2A,B*, the protocol described by Sparkes et al. was used (*Sparkes et al., 2006*). Briefly, Agrobacterium strains transformed with T-DNAs expressing the grown up overnight under Kanamycin and Gentamicin selection. These cultures were then pelleted and washed twice with infiltration media the next day and then resuspended at $OD_{600}$ 0.8 and infiltrated into the desired tissue using a needleless syringe. Imaging of the luminescence signal was performed between three and five days after infiltration, depending on the assay being performed.

## AgroBEST co-culture for transient expression of FBP

The transient expression assays for *S. lycopersicum* and *A. thaliana* were carried out using the AgroBEST protocol (*Wu et al., 2014*). Briefly, seeds were sterilized and germinated in 0.5x MS and once cotyledons emerged, the seedlings were co-cultured with the appropriate Agrobacterium strain that had been transformed with the T-DNA encoding an FBP, in a mixture of MS and AB-MES. For all tomato AgroBEST treatments FBP_6 was used (*Table 1*). For the *Arabidopsis* AgroBEST FBP_12 and FBP_11 (*Table 1*) were used. These strains were primed for AgroBEST the previous day through overnight culture in AB-MES salts according to the AgroBEST protocol. Visualization of the bioluminescence signal was performed after two days of co-culture.

## Generation of a stable transgenic FBP line of *Nicotiana benthamiana*

The protocol detailed in *Sparkes et al. (2006)* was used to generate stable lines of *Nicotiana benthamiana* with the FBP integrated into the genome. To summarize, the Agrobacterium strain characterized in *Figure 1* was used to infiltrate leaves of *N. benthamiana*. After three days these leaves were excised, surface sterilized and transferred to shooting media plates that contained Kanamycin for selection. After callus and shoot formation, the shoots were transformed to rooting media plates which also had Kanamycin (50 mg/L) in the media. Finally, rooted plantlets were screened for luminescence and the individual with the highest signal was transferred to soil for seed.

## CCD camera-based luminescence imaging

For all static CCD camera-based luciferase imaging eight-minute exposures of plant tissue were taken in using a UVP BioImaging Systems EpiChemi3 Darkroom. For transgenic plants a twelve-minute exposure was used. Paired bright field images were also taken using the same camera. The brightness and contrast of the long exposure images were adjusted using imageJ (*Schindelin et al., 2012*) to optimize signal visibility and then overlaid as a false colored image on the bright field image, where warmer colors correspond to higher signals.

## Time lapse CCD camera-based imaging

For the time lapse data collected in *Figure 3A,C* and in *Figure 4C*, experiments were performed using the NightOWL LB 983 in vivo imaging system (*Fenske et al., 2015*). In all cases Agrobacterium infiltration of the FBP into the desired tissue was performed as described previously. For the petunia time lapse imaging data displayed in *Figure 3A and C*, the petals of flowers were infiltrated and then the flowers were excised from the plants either one or two days after infiltrations and were mounted in the imaging platform with their pedicles immersed in a solution of 5% sucrose (*Fenske et al., 2015*). Images were then captured every hour with a ten-minute exposure. Long day light conditions (16 hr light/8 hr dark) were implemented in the times between images. For the data displayed in *Figure 4C*, leaves of *N. benthamiana* were infiltrated and then excised from the plant after three days and placed in petri dishes. For the watered conditions a moistened filter paper was added to the petri dish, whereas for the drought conditions the filter paper was left dry. Images were then captured every hour with a ten-minute exposure and the timepoint with the peak pRAB18:Luz signal was reported.

## Quantification of luminescence signal in images

For quantification of luminescence signal, imageJ (*Schindelin et al., 2012*) was used to box areas of the same size in images of infiltrated leaves or 96 well plates containing hole punches from infiltrated leaves and then the average signal intensity was recorded and plotted in python using the seaborn

(*Waskom et al., 2014*) plotting package. All p values reported were calculated using the t-test function in the scipy package. All the raw data and data analysis code are available on Github (https://github.com/craftyKraken/P4; *Chamness, 2020*; copy archived at https://github.com/elifesciences-publications/P4).

## Time lapse luminescence imaging using DSLR-based system

A complete listing of sourced components and costs is found in *Table 2*. Time-lapse images were captured using a Canon DSLR equipped with a macro lens and controlled from a Raspberry Pi microcomputer running a combination of free, open-source and custom software. The gphoto2 library was used to control the camera via USB; power to lights and the camera was modulated by a pair of controllable four outlet power relays connected to the Pi GPIO interface via jumper cables; a custom python wrapper was used to implement control logic for the camera, using gphoto2, and for the relay, using the RPi.GPIO package. Long-exposure images were captured in a dark room or light-proof pop-up tent. Image processing was performed using macros written in ImageJ, and the FFMPEG library, under top-level control of an additional custom python wrapper. Gphoto2, ImageJ and FFMPEG are free and open-source software; the custom wrappers are hosted on a public GitHub repository (https://github.com/craftyKraken/P4), and are also available for reuse under the GNU 3.0 public license.

## Acknowledgements

NL and TI are supported by grants from National Institute of Health (R01GM079712) and Next-Generation BioGreen 21 Program (PJ013386, Rural Development Administration, Republic of Korea). We would like to thank Dr. Karen Sarkisyan and Dr. Nadya Markina for kindly sharing the sequences of the FBP genes from their publication.

## Additional information

### Funding

| Funder | Grant reference number | Author |
| --- | --- | --- |
| University of Minnesota | Grand Challenges Postdoctoral Fellowship | Arjun Khakhar |
| National Institutes of Health | R01GM079712 | Nayoung Lee |
| Rural Development Administration | PJ013386 | Nayoung Lee |
| U.S. Department of Energy | DE-SC0018277 | Colby G Starker James C Chamness |

The funders had no role in study design, data collection and interpretation, or the decision to submit the work for publication.

### Author contributions

Arjun Khakhar, Conceptualization, Resources, Data curation, Software, Formal analysis, Supervision, Validation, Investigation, Visualization, Methodology, Project administration; Colby G Starker, Conceptualization, Validation, Investigation; James C Chamness, Investigation, Visualization; Nayoung Lee, Validation, Investigation; Sydney Stokke, Cecily Wang, Ryan Swanson, Furva Rizvi, Resources, Investigation; Takato Imaizumi, Daniel F Voytas, Supervision, Funding acquisition

### Author ORCIDs

Arjun Khakhar (iD) https://orcid.org/0000-0002-4676-6533
Colby G Starker (iD) http://orcid.org/0000-0002-6774-7227
Daniel F Voytas (iD) https://orcid.org/0000-0002-4944-1224

**Decision letter and Author response**
Decision letter https://doi.org/10.7554/eLife.52786.sa1
Author response https://doi.org/10.7554/eLife.52786.sa2

## Additional files

### Supplementary files
• Source code 1. Code for data analysis and plotting.

• Transparent reporting form

### Data availability

All the data collected for this study is depicted in the figures included in the manuscript. All the raw data used to make the figures has been uploaded.

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
