## [Decision Letter]

Thank you for submitting your work entitled "Building customizable auto-luminescent luciferase-based reporters in plants" for consideration by *eLife*. Your article has been reviewed by three peer reviewers, and the evaluation has been overseen by Richard Amasino as Reviewing Editor and Christian Hardtke as the Senior Editor.

First, the reasons we think this is an interesting Tools and Resources article. Bioluminance-based reporters have proven extremely useful for non-invasive and dynamic imaging of plant gene expression. One of the most powerful features of luminance reporters, as opposed to fluorescence reporters, is zero background. However, luminance reporters, such as luciferase, require a chemical substrate to generate light. Some substrates, such as D-luciferin are water soluble and can be applied to whole plants by watering, but these substrates are quite expensive, limiting the practicality of repeated imaging and longer-term dynamic studies. In addition, there can be questions about the contributions of differences in gene expression as opposed to differential substrate penetration in variation of signal strength. One solution to these challenges is to introduce a biochemical pathway to synthesize the luciferase substrate in planta from endogenous metabolites, has been done recently for the substrate of firefly luciferase in Nicotiana. However, this pathway requires transformation into plastids to be operational, which is challenging to do in most plants. Here the authors take advantage of a recently described fungal bioluminance pathway (FBP) that can be expressed in the nucleus and which utilizes a common plant metabolite, caffeic acid, to synthesize the substrate for the fungal luciferase *Luz*. They demonstrate that this complete pathway can be reconstituted in stable transgenic plants and can utilize native caffeic acid to generate luminance, improve the output by engineering a recycling pathway for the Lux substrate, show that the pathway can operate successfully in several plant species (most by transient expression), demonstrate utility of the tool for studying variance in gene expression over space and time, and report a versatile molecular toolkit for making luminance reporters. In addition, the authors develop and demonstrate a low-cost imaging platform utilizing consumer cameras and very inexpensive computational hardware for conducting bioluminance experiments, potentially enabling this technology for a wider range of users.

However, to be most useful to the scientific community there are several issues that need to be better addressed before we can accept your paper as discussed below.

You argue convincingly for the potential applications of your approach to luminescence imaging, but need to also clearly discuss the limitations and drawbacks-clarify what the technology will not be immediately useful for and the drawbacks of using this pathway as opposed to using exogenous luciferin as substrate for luciferase based reporter studies.

For example, you note that unlike luciferin delivery, which would not have even delivery across cells and tissues, your system will be in every cell and therefore be an improvement (Introduction). However, would not uneven distribution of the substrates for your luciferin biosynthetic cycle would lead to artefactual variation on top of any reporter dynamics being detected Also, uneven expression of the enzymes would lead to the same type of problems.

A complement to discussing possible limitations would be a more careful characterization of individual plants imaged with sufficient resolution to judge spatial heterogeneity. A time-lapse experiment to judge temporal heterogeneity in the stable transgenics would also be valuable.

Finally, regarding "Similar patterns of expression are observed in root tips of *Arabidopsis thaliana* with constitutively expressed firefly luciferase," you should compare your luminescence to that of published luciferase reporters provided with luciferin so if arguing similar patterns you can show the supporting results at higher resolution in the stable tobacco transgenics.

Related to the above point, although the ability to use endogenous caffeic acid is a boon for reducing costs and enabling longer and more detailed dynamic studies, there remain questions about the possible variance in caffeic acid availability over tissues and time, and how such variance might affect experimental outcomes and interpretation. What is known about how caffeic acid synthesis and turnover might vary over cells and tissues and how it might vary over time and physiological state? If substrate is limiting, such variation would be convolved with changes in reporter protein levels. Addition of substrate recycling helps assure higher substrate levels and is an important addition to the toolkit. We are not asking for more experiments on this issue, but for you to discuss in more depth the questions about possible variation in native caffeic acid levels and what future studies might be needed to assess the impact of such variation on interpreting reporter results.

Also, comment on how phylogenetically widespread caffeic acid production is in plants. You make the strong point that your system works in a range of species, but were those species chosen because they are known to make caffeic acid?

Analysis of dynamic patterns of gene expression are sensitive to the reporter half-life. Different luciferases for example can have widely different half-lives when expressed in vivo. It is not clear in this paper or in the 2018 Khotlobay paper what the relative stability of *Luz* might be. This is a relevant point for discussion in the paper.

The data showing an example gene expression analysis in Figure 3 are pretty scant, with and n of 1 for the pODO1 reporter and an n of 2 for the 34S promoter. The whiskers on the 35S data are stated to be standard deviation, but with an n of 2 reporting SD is not really valid. This experiment requires more repeats to be meaningful. The data in 3C have 7 repeats, which is far better, but do not show a control for the promoter used (like 35S). So both experiments are incomplete. It also seems to be the case that the amplitude of response shown here is much lower than what has been published using Luc. This should be commented on.

Subsection “Developing a low-cost, long-term bioluminescence imaging system” – "high resolution and detail for macroscopic plant subjects under multiple light conditions, and sufficient sensitivity to capture even low levels of bioluminescence signal." There is currently not sufficient spatial resolution to make this claim and no sensitivity testing compared with other methods.

[Editors' note: further revisions were suggested prior to acceptance, as described below.]

Thank you for submitting your revised article "Building customizable auto-luminescent luciferase-based reporters in plants" for consideration by *eLife*. Your revised article has been re-read by two peer reviewers, and the evaluation has been overseen by Rick Amasino as the Reviewing Editor and Christian Hardtke as the Senior Editor.

The reviewers have discussed the reviews with one another and the Reviewing Editor has drafted this decision to help you prepare a revised submission.

We are still concerned that Figure 3 does not strongly support the contention that there is significant temporal variation in expression as compared to controls. The curves for Figure 3C and 3D appear to be subtly different, but it is difficult to assess the significance. Although you have now provided the 35S controls for 4C, there is no statistical support for comparing the trends in the experimental versus control, just plots of means over time and their SDs. Are these curves really different? If there is uncertainty (we appreciate that there is noise in a transient experiment like this), then address that in the text.

---

## [Author Response]

First, the reasons we think this is an interesting Tools and Resources article. Bioluminance-based reporters have proven extremely useful for non-invasive and dynamic imaging of plant gene expression. One of the most powerful features of luminance reporters, as opposed to fluorescence reporters, is zero background. However, luminance reporters, such as luciferase, require a chemical substrate to generate light. Some substrates, such as D-luciferin are water soluble and can be applied to whole plants by watering, but these substrates are quite expensive, limiting the practicality of repeated imaging and longer-term dynamic studies. In addition, there can be questions about the contributions of differences in gene expression as opposed to differential substrate penetration in variation of signal strength. One solution to these challenges is to introduce a biochemical pathway to synthesize the luciferase substrate in planta from endogenous metabolites, has been done recently for the substrate of firefly luciferase in Nicotiana. However, this pathway requires transformation into plastids to be operational, which is challenging to do in most plants. Here the authors take advantage of a recently described fungal bioluminance pathway (FBP) that can be expressed in the nucleus and which utilizes a common plant metabolite, caffeic acid, to synthesize the substrate for the fungal luciferase Luz. They demonstrate that this complete pathway can be reconstituted in stable transgenic plants and can utilize native caffeic acid to generate luminance, improve the output by engineering a recycling pathway for the Lux substrate, show that the pathway can operate successfully in several plant species (most by transient expression), demonstrate utility of the tool for studying variance in gene expression over space and time, and report a versatile molecular toolkit for making luminance reporters. In addition, the authors develop and demonstrate a low-cost imaging platform utilizing consumer cameras and very inexpensive computational hardware for conducting bioluminance experiments, potentially enabling this technology for a wider range of users.However, to be most useful to the scientific community there are several issues that need to be better addressed before we can accept your paper as discussed below.You argue convincingly for the potential applications of your approach to luminescence imaging, but need to also clearly discuss the limitations and drawbacks-clarify what the technology will not be immediately useful for and the drawbacks of using this pathway as opposed to using exogenous luciferin as substrate for luciferase based reporter studies.For example, you note that unlike luciferin delivery, which would not have even delivery across cells and tissues, your system will be in every cell and therefore be an improvement (Introduction). However, would not uneven distribution of the substrates for your luciferin biosynthetic cycle would lead to artefactual variation on top of any reporter dynamics being detected Also, uneven expression of the enzymes would lead to the same type of problems.

To more thoroughly discuss the limitations of FBP reporters, specifically the lower signal strength and potentially variability of relying on an endogenously synthesized luciferin substrate, we added the following lines:

“Compared to these previously published luciferase reporters, which are typically provided an excess of substrate during application, we observe a lower amplitude response with our FBP reporters, which rely on an endogenously synthesized luciferin substrate.”

“However, caffeic acid levels may vary between tissues and environmental conditions^29^ This variation could complicate deployment of FBP reporters for gene expression by requiring an additional constitutive expression control.”

We also have added the characterization of a caffeic acid biosynthesis pathway in an autoluminescent line of *benthamiana* to the Results section and highlight how this might be an avenue to address these limitations in the future.

A complement to discussing possible limitations would be a more careful characterization of individual plants imaged with sufficient resolution to judge spatial heterogeneity. A time-lapse experiment to judge temporal heterogeneity in the stable transgenics would also be valuable.

We also replaced the data presented in Figure 1 F,G with images of stable T2 transgenic lines at sufficient resolution to descript the spatial and developmental heterogeneity of the luminescence signal. “We saw stronger signals in the root tips, shoot apical meristem, consistent with a higher density of cells in these tissues. Younger leaves also seem to have lower luminescence than older leaves in the young plants we imaged (Figure 1 F,G).”

Finally, regarding "Similar patterns of expression are observed in root tips of *Arabidopsis thaliana* with constitutively expressed firefly luciferase," you should compare your luminescence to that of published luciferase reporters provided with luciferin so if arguing similar patterns you can show the supporting results at higher resolution in the stable tobacco transgenics.

We removed the claim made as collecting the data to definitely make it is beyond the scope of this manuscript.

Related to the above point, although the ability to use endogenous caffeic acid is a boon for reducing costs and enabling longer and more detailed dynamic studies, there remain questions about the possible variance in caffeic acid availability over tissues and time, and how such variance might affect experimental outcomes and interpretation.

We added the section on caffeic acid biosynthesis to address the reviewer concerns about the heterogeneity confounding the results from FBP reporters of gene expression. We highlight this potential source of variation, “However, caffeic acid levels may vary between tissues and environmental conditions^29^.”, and go on to propose a potential solution for future applications of this reporters if such issues arise, “In the future this pathway might be used in conjunction with the FBP to create reporters that are more robust against natural variation in caffeic acid biosynthesis.”

What is known about how caffeic acid synthesis and turnover might vary over cells and tissues and how it might vary over time and physiological state? If substrate is limiting, such variation would be convolved with changes in reporter protein levels. Addition of substrate recycling helps assure higher substrate levels and is an important addition to the toolkit. We are not asking for more experiments on this issue, but for you to discuss in more depth the questions about possible variation in native caffeic acid levels and what future studies might be needed to assess the impact of such variation on interpreting reporter results.Also, comment on how phylogenetically widespread caffeic acid production is in plants. You make the strong point that your system works in a range of species, but were those species chosen because they are known to make caffeic acid?

We added the basis for broad phylogenetic conservation of caffeic acid biosynthesis, “Caffeic acid is a requisite intermediate of lignin biosynthesis and is thus ubiquitous in higher plants^32^”. We also included a rationale of why we chose the species we did in Figure 2, “To test this hypothesis, we tested the FBP in multiple plant species that were amenable to Agrobacterium infiltration.”

Analysis of dynamic patterns of gene expression are sensitive to the reporter half-life. Different luciferases for example can have widely different half-lives when expressed in vivo. It is not clear in this paper or in the 2018 Khotlobay paper what the relative stability of Luz might be. This is a relevant point for discussion in the paper.

We address the stability of *Luz* and the ramifications on the temporal resolution of FBP reporters “Further characterization of the stability of the *Luz* protein will be needed to determine the temporal resolution of FBP-based reporters. However, the data presented here do show that our FBP reporters can be used to visualize changes in gene expression that occur on the timescale of hours.”

The data showing an example gene expression analysis in Figure 3 are pretty scant, with and n of 1 for the pODO1 reporter and an n of 2 for the 34S promoter. The whiskers on the 35S data are stated to be standard deviation, but with an n of 2 reporting SD is not really valid. This experiment requires more repeats to be meaningful. The data in 3C have 7 repeats, which is far better, but do not show a control for the promoter used (like 35S). So both experiments are incomplete. It also seems to be the case that the amplitude of response shown here is much lower than what has been published using Luc. This should be commented on.

To address the reviewer concerns for Figure 3 we specify that Figure 3A is a representative example, included for illustrative purposes, and removed the erroneous mention of standard deviation. Figure 3 legend – “A) A representative example of time course luminescence data from long day entrained *P. hybrida* flowers infiltrated with FBPs. The orange line represents FBP with *Luz* driven by the *ODO1* promoter from Petunia and the blue line represents the mean luminescence signal of an FBP with *Luz* driven by the 35S promoter”. We also added 35S controls for Figure 3C. We comment on the lower observed Luc signal “Compared to these previously published luciferase reporters, which are typically provided an excess of substrate during application, we observe a lower amplitude response with our FBP reporters, which rely on an endogenously synthesized luciferin substrate.”

Subsection “Developing a low-cost, long-term bioluminescence imaging system” – "high resolution and detail for macroscopic plant subjects under multiple light conditions, and sufficient sensitivity to capture even low levels of bioluminescence signal." There is currently not sufficient spatial resolution to make this claim and no sensitivity testing compared with other methods.

We changed the wording to make our claim more clear: “high resolution and detail for macroscopic plant subjects under multiple light conditions, and sufficient sensitivity to capture the bioluminescence signal from stable or transient expression of the FBP”.

[Editors' note: further revisions were suggested prior to acceptance, as described below.]

The reviewers have discussed the reviews with one another and the Reviewing Editor has drafted this decision to help you prepare a revised submission.We are still concerned that Figure 3 does not strongly support the contention that there is significant temporal variation in expression as compared to controls. The curves for Figure 3C and 3D appear to be subtly different, but it is difficult to assess the significance. Although you have now provided the 35S controls for 4C, there is no statistical support for comparing the trends in the experimental versus control, just plots of means over time and their SDs. Are these curves really different? If there is uncertainty (we appreciate that there is noise in a transient experiment like this), then address that in the text.

We appreciate the reviewer’s comment that the time course plots do not do enough to highlight the differences between the pODO1 and p35S FBPs. To more quantitatively assess whether the luminescence derived from the pODO1*:Luz* construct peaks differently than that of the p35S*:Luz* control and whether the temporal dynamic of the pODO1*:Luz* signal is significantly different from the p35S one, we have added a panel to Figure 3, Figure 3 D. Here we plot the time of peak luminescence for all the time courses shown and, based on a t-test, demonstrate statistically significant differences between the temporal dynamics of the pODO1 and p35S constructs at 2 days post infiltration to make this point clear.